# Principle and Method for Determining the Calendar Safety Life of Aircraft Structural Protection Systems

Teng Zhang [1], Tianyu Zhang [1,*], Yuting He [1], Bo Hou [2] and Changfan Li [2]

1   Aeronautical Engineering College, Air Force Engineering University, Xi'an 710038, China; zt_gm@126.com (T.Z.)
2   Research Institute of Army Aviation, Beijing 101100, China
*   Correspondence: zz664191970@163.com

**Abstract:** The calendar safety life of the surface protection system in aircraft structures is the time limit for it to be used without functional failure at a certain level of reliability and confidence. The reliability of such protection systems and the operational safety and economy of the structure are closely related. This paper firstly establishes two methods for determining the calendar safety life of aircraft structural protection systems under a single service environment and in multiple service environments. A method for determining the reliability of the calendar safety life of the aircraft structural protection system was proposed, and an expression of the relationship between the maintenance costs for the aircraft fleet and the reliability of the calendar safety life of the aircraft structural protection system based on the relationship between the amount of corrosion damage to the structural substrate and the corrosion time and the expression of the calendar safety life of the protection system was established. Finally, taking a hypothetical aircraft fuselage wall plate connection structure as an example, an alternating corrosion fatigue test with protection system specimens was carried out. The process for determining the calendar safety life of the structural protection system and its reliability are given. This method is important to ensure the safety of aircraft structures, improve the efficiency of use, and reduce maintenance costs. Generally speaking, the reliability of the calendar safety life of the structure is 99.9%, and after the analysis in this paper, the reliability of the structural protection system is about 70%.

**Keywords:** principal; method; protection system; calendar safety life; aircraft structure

## 1. Introduction

Protective systems, such as coatings applied on the surface of aircraft structures, can effectively protect the structure from corrosion and ensure the safe flight of the aircraft [1]. Surface coating systems play a very important protective role for aircraft performing cruise missions in severe weather areas (e.g., high temperature, high humidity, high salt, high sand, strong UV, etc.) [2,3]. The complete protective system for aircraft structures consists of an anodic oxide film, primer, intermediate and topcoat [2]. However, protective coatings are used in many cases as single or separate coating systems, which are related to the specific service environment and operating conditions. Moreover, the correct combination for the pre-treatment method of the substrate and the protective coating system determines the corrosion resistance of the whole protection system. Aircraft that are often in service in coastal areas, island areas or industrial areas are subject to harsh environmental corrosion [4–7]. Moreover, because the aircraft is a special closed structure, water often accumulates inside it and causes corrosion, such as in the seam area between the fuselage and the wing skin, the connection area between the bolts or rivets and the plates, the landing gear of the aircraft, the area where the battery is placed, the recess area between the flaps and the hinges where water easily accumulates, the area affected by the engine exhaust, and other places where water easily accumulates [8]. The main substrate materials

studied are metallic materials, such as aluminum alloys, and the main coating materials studied are organic coatings, such as epoxy primers. Because the main object in this paper is the aircraft structure, most of the fuselage surfaces, landing gear, and various connectors in the aircraft are aluminum alloy materials; moreover, the aluminum alloy materials used in current civil aircraft structures in China account for about four-fifths of all materials, so aluminum-based materials are used in this paper.

The calendar safety life of the surface protection system in an aircraft structure is the time limit for its use without functional failure at a certain level of reliability and confidence [1]. Our research is focused on the force-bearing components of aircraft structures, and the main object of our research is the metallic structures and coating systems for aircraft surfaces. For critical structures affecting flight safety, and for structures with particularly poor corrosion resistance regarding the substrate materials (e.g., landing gear), the calendar safety life of protection systems should be selected at a high level of reliability and confidence based on the safety of their use [2]. For other aircraft structures, since it still takes a certain amount of time for the corrosion failure of the structural substrate to occur after the failure of the protection system, structural safety, overhaul time, and repair costs, among other factors, should be considered comprehensively [3]. Repair costs and other factors should be considered from an economical point of view to determine the reliability and confidence of the protection system [4]. Therefore, the calendar safety life of the protection system for these structures is mainly related to the use of economics [5].

There are many types of surface protection systems for aircraft structures, such as topcoats, primers, corrosion inhibitors, anodic oxidation layers, galvanized layers, and aluminum cladding layers [6]. The various types of protection systems can be divided into many different grades [7]. When the surface protection system for the structure is damaged after the aircraft has been in service for a period, some types of protection systems can be repaired during a structural overhaul, such as the topcoat and primer, while others cannot, such as the anodic oxide layer and the aluminum cladding layer [8]. If the surface protection system has corroded to the inner layer before the first overhaul (first flip), causing irreparable damage to the protection layer, the state of the protection system after the first flip is worse than when the aircraft leaves the factory, and its calendar safety life is shorter than the calendar safety life of the protection system before the first flip [9].

Therefore, the calendar safety life of the protection system is divided into two categories: the calendar safety life of the structural protection system before the first overturn, and that after the first overturn. The former is the calendar life limit with a specific probability of corrosion failure, corresponding to the state of the structural protection system after the aircraft leaves the factory. This is mainly related to the determination of the first overturn period of the structure. The calendar safety life of the structural protection system after the first overturn is the calendar life limit with a specific probability of corrosion failure corresponding to the state of the structural protection system after the aircraft is overhauled. The calendar life limit for the probability of failure of a structural protection system after the first overhaul is mainly related to the determination of the structural overhaul interval. It should be noted here that the difference between the calendar safety life for the protection system before the first flip and after the first flip is whether the inner protection system is damaged. If the inner protection system is intact at the time of the first flip, the protection system can be repaired to factory conditions at the time of the first flip. Then, the calendar safety life of the structure after the first flip is still managed based on the calendar safety life before the first flip.

This paper presents a principle and method for determining the calendar safety life of an aircraft structural protection system, and proposes a method for determining its reliability and, finally, validates and analyzes this method by means of an example. Failure of a protective coating on an aircraft structure does not necessarily mean that the structure itself has failed. The structure will be selected with a certain reliability in the life determination, and current common practice is to select a reliability level of 50%. This means that the maintenance cost for the aircraft is too high. The research in this paper found

that the reliability of the aircraft structural protection system can meet the requirements at about 70%, which means that this can greatly reduce the maintenance costs and be more economical, while meeting the safety requirements.

## 2. Principles and Methods for Determining the Calendar Safety Life

### 2.1. Single Service Environment

The calendar safety life of an aircraft structural protection system is based on the number of years it has been in service [1]. If an aircraft is in a similar environment concerning the service area during its lifetime, it can be considered that the aircraft is in service in a single environment [2–4]. As such, an annual corrosion spectrum for the service area, where the aircraft is located, is the only thing needed to prepare [3,4]. Then, the calendar safety life of the aircraft structural protection system can be obtained through testing and analysis. If the aircraft is in service in different areas with large differences in the environment during its lifetime, it is considered that the aircraft is in service in multiple environments. In this case, an annual corrosion spectrum for the different areas for testing and analysis to obtain the calendar safety life of the structural protection system is needed to prepare [6,8].

The calendar safety life of a surface protection system is related to the materials and processes selected for the protection system, as well as the environment and service history of the aircraft, the location of the structure, and the reliability and confidence level selected according to the service requirements [1,3–5]. The first three parameters determine the real failure time for the protection system. The selected reliability and confidence levels determine the safety level for the calendar safety life of the protection system [6–8]. The reliability and confidence levels for the calendar safety life of the protection system should be determined according to the importance of the structure, maintenance costs, and other comprehensive considerations [8,9].

For the critical load-bearing parts in an aircraft structure or when the structural base material corrosion resistance is particularly poor, a higher reliability level should be selected to ensure that the probability of the protection system failing before the structural overhaul is very low [3,10]. This ensures that the pre-overhaul base structure does not fail due to corrosion [10]. For general structures, the reliability level should be determined mainly from the perspective of the economy of the structural use and maintenance [7]. The level should not be too high, such that the structure is frequently repaired, affecting the equipment integrity rate and increasing the maintenance costs. Nor should it be too low, such that the substrate corrosion becomes overly serious, leading to major repairs to the structure or even replacement parts, which also increases the maintenance costs. In addition, the selection of the calendar safety life reliability for the structural protection system must ensure that the structural matrix for this reliability corresponds to the calendar cycle without fracture failures due to the effects of corrosion [7–9,11].

Since determining the calendar life of a structure using the actual service conditions requires a long test cycle, corrosion problems are generally investigated in engineering by conducting equivalent tests under accelerated laboratory conditions [12,13]. The technical approach to determine the calendar safety life of a surface protection system is shown in Figure 1. Several major tasks involved in this process are described as follows [4].

(1)    Designing and manufacturing simulated specimens for structural protection systems

The object of the study of the calendar safety life of a protection system should be selected according to the purpose of the study. For example, to determine the calendar safety life of the aircraft fuselage, the main components most vulnerable to corrosion are usually selected as the object of the study [14].

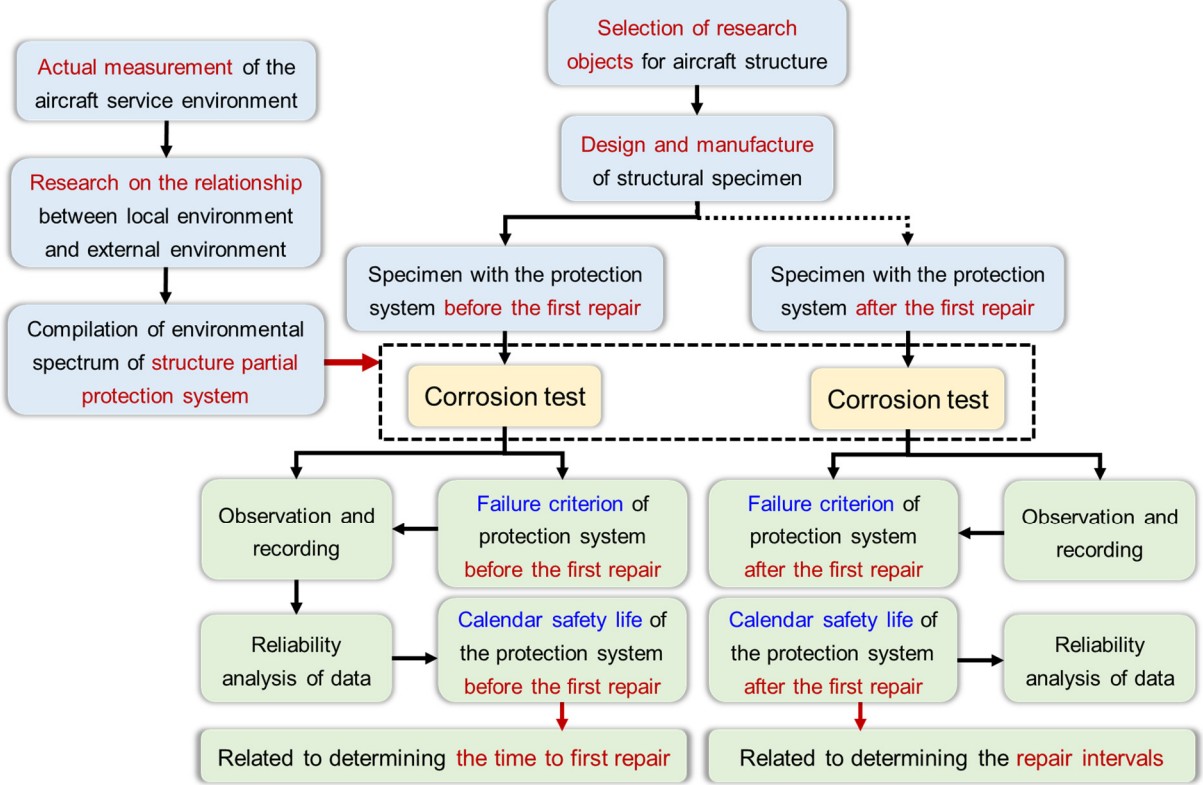

**Figure 1.** Technical ways to determine the safety life of protective systems.

The design of the protection system simulation specimens follows three principles [1–4]: (1) The specimens should use the same surface protection system materials as the actual structure of the aircraft, including the same base material, the same type of protection system and material grades, the same structure processing technology, and the same thickness of the protection layer. (2). The specimens should have the same local characteristics as the actual structure, such as the study object and the other structures connected through rivets. The specimens should be considered for the design of the contact parts, such as the actual structure of the existence of tabs, round holes, and other features affecting the corrosion patterns of the protection system. (3). The production of a fleet of aircraft requires a long time, so the actual fleet of protection system materials used may not be batch produced. For specimen processing, different batches of the same grade of protection system materials should be chosen where possible to reflect the dispersion of the material.

When producing mockups, the first mockup of the pre-rollover structure should be produced based on the state of the aircraft structure at the factory [15]. When determining the failure criterion for the protection system before the first overturning of the structure (i.e., the possible state of the surface protection system at the first overturning) based on this simulated part, the post-first overturning structural simulated part can be produced based on the criterion [16]. For example, if corrosion of the structural cladding occurs as the failure criterion for the structural mockup before the first overturning, the actual structure of the aircraft managed according to this calendar safety life will have some areas where cladding may fail during the overhaul. Since the cladding and the anodized layer outside the cladding are not repaired during the overhaul of the aircraft structure, generally only the primer and topcoat are repainted [17]. Therefore, in order to more reasonably simulate the state of the post-first overturn protection system, it is necessary to produce a mockup of the surface protection system. To more reasonably simulate the state of the post-first flip protection system and from the perspective of partial safety, the designed post-first flip structure's simulation parts should be sprayed with primer and topcoat only, without aluminum cladding and anodizing treatment.

(2)　Preparation of the environmental spectrum for local structural protection systems

To conduct accelerated test research under laboratory conditions, it is necessary to ensure that the accelerated environmental spectrum used in the test is equivalent to the corrosion damage effect of the real service environment for the aircraft structure. This requires the preparation of a local accelerated environmental spectrum for the part where the structure is located.

First, the actual service environment of the aircraft is considered in order to carry out real measurements to determine the external environmental characteristics of the aircraft. Second, the local environment of the structure to be studied and the relationship between the external environment model are established. Again, based on the determined local environmental characteristics of the aircraft structure, the structural accelerated environment spectrum is prepared using existing methods [6–8].

(3)　Judging the first flip before and the first flip after the failure of the protection system

The number of load cycles corresponding to the fracture of the structure during the fatigue test is the fatigue life. However, the study of the calendar life differs from this. The criteria for determining the calendar life and the life of the protection system are often not visually expressed. Therefore, to evaluate the calendar safety life of the protection system, it is necessary to give the criteria for the calendar life of the protection system. The failure of the protection system will cause corrosion to the structural substrate, so the failure of the protection system before the first overturning can be obtained by observing whether corrosion damage occurs to the structural substrate. Similarly, the failure of the protection system after the first overturning can be obtained by observing whether irreparable damage occurs to the inner protection system.

To obtain the specific process for the failure of the protection system, different cycles of corrosion tests to record the damage characteristics of the protective layer and the specific location of the damage must be carried out first. Then, through microscopic observation, energy spectrum analysis is conducted, and other means are used to determine whether the structural substrate (or the irreparable protective layer) is corroded. The specific location of the corrosion is, thus, recorded. Finally, through the location of the damage to the protective layer and the substrate (or the irreparable protective layer), the corrosion position of the corresponding relationship is used to find the corrosion of the substrate based on the corresponding damage characteristics of the protective layer. This is conducted to obtain the criteria for the pre-flip (or post-flip) failure of the protective system.

(4)　Carry out corrosion tests on the simulated parts of different protection systems

In the preparation of the accelerated environmental spectrum to carry out the first turn over before and after the first turn over, the protection system simulation of the corrosion test takes place. In the test process, the equivalent corrosion spectrum is applied after a cycle (e.g., equivalent to a year) to a specimen for surface observation and recording. When the damage characteristics of the specimen of the surface protection system corresponds to the failure criterion, the specimen of the surface protection system reaches the calendar life limit in equivalent corrosion years.

From the perspective of saving costs, the corrosion test for the first post-turn over structural simulator can be omitted. As the protection system for the pre-flip structure mockup is better than that of the post-flip structure mockup, the failure criterion of the post-flip structure mockup usually appears before the failure criterion of the pre-flip structure mockup. (For example, damage to the paint layer often appears before damage to the aluminum cladding layer.) Therefore, only the corrosion test for the structural simulator before the first flip can be carried out. Each specimen is recorded twice when the above two judgments appear, and the calendar time corresponding to the failure judgment for the structural simulator after the first flip is used as the calendar life of the structural protection system after the first flip.

(5)  Reliability analysis of the test data and the determination for the calendar safety life of the protection system

From the perspective of saving test funds, test parts are usually only made into simulated parts that reflect the characteristics of the actual structure, rather than the actual structural parts. Therefore, the surface area of the structure of the specimen is generally only a fraction of the surface area of the actual structure. Assuming the actual structure in the key parts of the surface area for the specimen of the key parts of the surface area is $k$ times, the actual structure in the service environment can be equivalently seen as $k$ specimens, while accepting corrosion. Any one of the surface protection system failures is to be regarded as the entire actual structure of the surface protection system failure. Therefore, the reliability of the specimen selected, and the reliability of the actual structure are different. When studying the calendar safety life of the surface protection system of an actual structure, it is necessary to take into account the difference in surface area between the specimen and the actual structural piece in critical areas.

The calendar safety life reliability for the surface protection system of an actual structure is chosen to be $\alpha$. Then, for a sample of the actual structure with sample capacity $n$, there are approximately $n(1 - \alpha)$ structures, with the surface protection system failing when the calendar safety life of the protection system is reached. From a the point of view of safety, for a sample of a specimen with sample capacity $nk$ (the surface area of the actual structure is $k$ times the surface area of the specimen; then, a sample of the specimen with sample capacity $nk$ is equivalent to a sample of the actual structure with sample capacity $n$) it is still required that $n(1 - \alpha)$ specimens of the surface protection system of the specimen fail when the calendar safety life of the protection system is reached. Assuming that the calendar safety life reliability should be selected as $\alpha'$ for the calendar safety life analysis of the surface protection system of the actual structure with the test data, the relation $nk(1 - \alpha') = n(1 - \alpha)$ can be obtained [1] as follows:

$$\alpha' = (k + \alpha - 1)/k \tag{1}$$

The analysis results obtained from the test data according to reliability $\alpha'$ can be used as the calendar safety life of the protection system for the actual structure at reliability $\alpha$. The above issues are further elaborated below with the help of Figure 2.

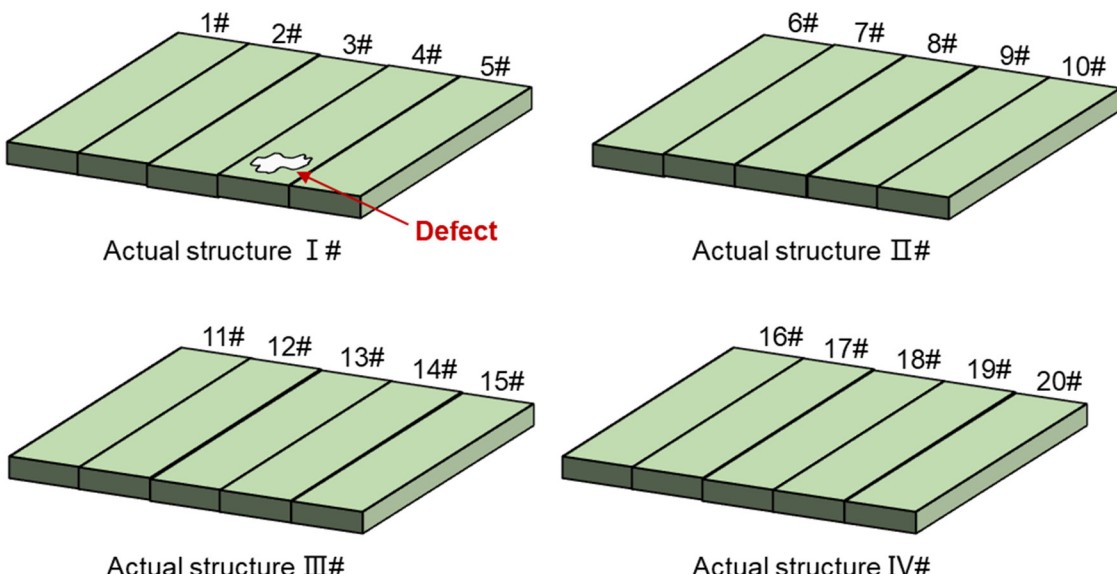

**Figure 2.** Schematic diagram of the proportional relationship between the specimen and the actual structure.

Figure 2 features four pieces of actual structure, with serial numbers from I# to IV#. Assuming that the surface area of the actual structure is five times that of the specimens, the sum of the surface area of the four pieces of the actual structure is equivalent to the surface area of 20 specimens. The serial numbers for the specimens are 1# to 20#. Then, it is assumed that the calendar safety life of the actual structure surface protection system is 10 years, and the reliability of the calendar safety life is 75%. This means that when four structures are in service/use for 10 years, about $4 \times (1\%\text{--}75\%) = 1$ structure surface protection system failure occurs, such as the defect in the I# structure. From a safety perspective, for a sample capacity of $4 \times 5 = 20$ specimen samples, when the calendar safety life of the protection system is 10 years, at most one specimen of the surface protection system is still required to fail, such as the defect in the 4# specimen. Then, the specimen surface protection system calendar safety life reliability is $(20 - 1)/20 = 95\%$. It can be seen that, because the surface area of the specimen is smaller than the actual structure, the reliability of the specimen surface protection system calendar safety life is obviously greater than the actual structure. The analysis results obtained from the test data, based on the above reliability, can be used as the calendar safety life of the protection system for the actual structure at a reliability of $\alpha$. With the obtained test results and the given reliability, the calendar safety life of the structural protection system should be determined by first determining the distribution of the calendar safety life of the protection system. Then, the calendar safety life value that satisfies the given reliability and confidence levels according to the probability density function of the determined distribution were obtained. Since there is a general method for determining the form of the distribution and calculating the reliability value, it is not repeated here.

### 2.2. Multiple Service Environments

Aircraft are usually in service in multiple regions during their full life service [9]. Civil aircraft usually switch between multiple cities during the routes flown. Military aircraft also often switch between regions [10]. Due to the differences in the corrosive environments in different regions, the calendar safety life of the structural protection system determined by the equivalent environmental spectrum in one region alone cannot be used to manage the calendar life of the protection system for aircraft in service in multiple regions [11].

Existing studies have shown that the damage effect of corrosion on materials can be considered to obey a linear law [10–12]. On this basis, the determination for the calendar safety life of aircraft structural protection systems in a multiservice environment can be carried out by calculating the linear cumulative damage, as follows:

(1)　Preparation of the local environmental spectrum for different service areas

According to the climatic environment parameters for different service areas and the selected research objects, the local environmental spectrum for the protection system corresponding to the aircraft in service in different areas is prepared [13]. For example, the aircraft will be in service in four areas, A, B, C, and D. Among them, the service environments in A and B are similar, so a total of three environmental spectra need to be compiled, as follows: (1) the environmental spectrum corresponding to A and B, (2) the environmental spectrum corresponding to C, and (3) the environmental spectrum corresponding to D.

(2)　Determining the calendar safety life of the protection system under different service environments

According to the process shown in Figure 1, the tests and analyses on the protection system under different environmental spectra are carried out separately to determine the calendar safety life of the protection system under different environmental spectra [14,15]. For example, through testing and analysis, the calendar safety life of the protection system before and after the first overturning under the environmental spectrum: (1) is 10 years and 9 years, respectively; the calendar safety life before and after the first overturning under environmental spectrum (2) is 5 years and 4 years, respectively; and the calendar safety

life before and after the first overturning under environmental spectrum (3) is 7 years and 6 years, respectively.

(3)   Calendar safety life prediction based on the usage plan

According to the calculation method for linear cumulative damage, if the calendar safety life of the protection system in a certain environmental spectrum is $T$ years, and the aircraft is in service/use in its corresponding area for $t$ years ($t < T$), then the degree of damage to the protection system in these $t$ years is $t/T$. When the cumulative degree of damage to the protection system reaches 1, the protection system is considered to have reached the expected safe use cycle limit, and the sum of the corresponding calendar life at this time is the multi-service area protection system calendar safety life value.

According to the above example, a new aircraft is scheduled to be in service for 5 years at site A, 3 years at site B, 5 years at site C, and 7 years at site D. According to the cumulative damage calculation, the cumulative damage before the first rollover of the protection system for 5 years of service at A and 3 years of service at B is 0.8. In other words, the aircraft protection system will need to be repaired after 1 year of service, at most, in site C. After the first flip, the protection system will continue to serve in site C for 4 years, and the accumulated damage will reach 1 after the first flip. Then, it needs to be repaired for the second time. The aircraft will be transferred to service in site D after the overhaul, and the protection system will need to be repaired for the third time after 6 years of service. After the aircraft is overhauled and transferred to site D, a third repair is required after 6 years of service, and the repaired protection system can be safely used in the remaining 1 year of the service cycle.

It should be noted that the first turn over period and overhaul interval of the aircraft structure are usually determined based on the fatigue safety life of the critical load-bearing structure [16,17]. The calendar safety life of the protection system is only related to the process for developing the first turn over period and overhaul interval, rather than playing a decisive role. When the aircraft structure reaches the overhaul period and the protection system has not yet reached its service limit, the protection system is usually also repaired. By contrast, when the protection system reaches its service limit and the aircraft has not yet reached the overhaul period, a comprehensive analysis based on damage to the protection system and the calendar safety life of the structural substrate is required to decide whether to advance the structural overhaul.

## 3. Principles and Methods for Determining the Calendar Safety Life Reliability

The fatigue safety life at this stage of developing the service and life limits for aircraft structures does not refer to the life of the aircraft structure at the time of damage, but rather to a service/life cycle limit value for an aircraft with high reliability [1–3]. Since structural damage can be a direct threat to flight safety, its reliability level is generally high. For example, if the fatigue life of an aircraft structure obeys a lognormal distribution, it needs to satisfy a 99.9% reliability level with 90% confidence [13,16–18]. However, for the protection system in an aircraft structure, its failure does not immediately lead to structural damage, and a suitable reliability level can be found in determining its calendar safety life, which can ensure structural safety and improve the economy of use of the aircraft structure at the same time.

The calendar safety life of an aircraft structural protection system is not only related to the protection system itself and the environment in which the aircraft is used, but also to the selected reliability [16–19]. If the reliability chosen is too low, more structural protection systems may fail when the calendar safety life of the protection system is reached, resulting in the corrosion of more structural substrate materials [20–22]. As such, more parts will need to be repaired during the aircraft overhaul, and some parts may not be repaired in time, due to the failure of the protection system, resulting in the replacement of the structure [23,24]. This increases aircraft maintenance costs. If the reliability selected is too high, it will lead to a short calendar safety life of the protection system [12,25]. This will make structural overhauls more frequent, not only increasing the inspection and maintenance costs of the

aircraft, but also affecting the combat readiness rate of the aircraft [3–5,26]. In addition, the selection of the calendar life reliability for the structural protection system must ensure that the structural substrate does not fail within the calendar period corresponding to this reliability [1–3,27,28].

Therefore, it is important to select a suitable reliability level for the calendar life of the aircraft structure protection system, to ensure the safety of the aircraft structure, improve the efficiency of use, and reduce maintenance costs [3,5,29].

The selection of the reliability level for the calendar life of the aircraft structural protection system is closely related to the corrosion damage law for the structural substrate [28–31]. The selected reliability level should not only ensure the safety of the structure, but also optimize the costs of aircraft maintenance.

The protection system's calendar safety life reliability is determined using the following methods [1,2].

(1) Establish the relationship between the amount of structural substrate corrosion damage and the corrosion time model

In a corrosive environment, damage to the aircraft structure occurs in a variety of ways: an unloaded structure may suffer from pitting, spalling, etc., or a loaded structure may suffer from corrosion fatigue, stress corrosion cracking, etc.

According to the type of corrosion damage to the structural substrate, accelerated corrosion tests are conducted based on a structural substrate simulator in laboratory conditions. The test process requires a specimen without a surface protection system and one that can reflect the local characteristics of the structural substrate. The test conditions (the corrosion environment and load environment) can reflect the actual service environment of the aircraft structure and have a clear equivalence relationship. The specimens are divided into several groups. Each group undergoes different corrosion cycles. The amount of structural corrosion damage after different corrosion cycles were determined and the average amount of corrosion damage to the structural substrate and corrosion time relationship model were established [1–3,5] as follows:

$$h = f(t) \qquad (2)$$

In the above formula, $h$ is the average amount of corrosion damage, that is, the reliability of 50% of the corrosion damage; $t$ is the equivalent corrosion time; and $f(t)$ is the average amount of corrosion damage as a function of the corrosion time of the substrate.

When the amount of structural damage reaches $h_C$, it is considered that structural repair is no longer economical, or that the repaired structure cannot guarantee flight safety and structural replacement is required. When the amount of structural damage reaches $h_B$, it is considered that the structure can no longer be used, owing to, for example, the risk of a sudden fracture.

An example was provided to explain the replacement critical damage value $h_C$ and the scrap critical damage value $h_B$ of an aircraft structure. The structure of an aircraft is a critical part of damage tolerance. The critical crack length in the structure in service/use load is 200 mm, but when the crack length exceeds 150 mm, damage to the structure is very serious and can no longer ensure flight safety. The structure must be stopped from being used further. That is, a 150 mm crack length is the critical damage value for the structure for scrapping $h_B$. When the length of the structure does not exceed 50 mm, the structure can be reamed to stop holes and cracking, or reinforcements and other measures can be used for the repair. Once the crack length exceeds 50 mm, however, the repair cost of this structure exceeds the cost of the replacement parts, and repair is no longer economical. That is, there is a 50 mm crack length in the structure of the replacement at the critical damage value $h_C$. For $h_B$, when the structural damage is greater than $h_C$ and less than $h_B$, the structure can continue to maintain safe service, but the repair is uneconomical.

(2)    Carry out a corrosion test on the protective system specimens to establish the expression of the calendar safety life of the protective system

A corrosion test on the specimen of the protection system is carried out in an equivalent accelerated corrosion environment, according to the protection system calendar life of the failure judgment. This is conducted to derive the equivalent corrosion time when different specimens reach the failure judgment, that is, the calendar life of different specimens.

According to the test results, the distribution law on the calendar life of the protection system was determined and the expression of the calendar safety life of the protection system was derived as follows [4–6]:

$$N_\alpha = \overline{N} - k_\alpha \cdot S \tag{3}$$

In the above equation, $N_\alpha$ is the calendar safety life of the protection system under reliability $\alpha$; $\overline{NX_T}\overline{X_T}$ is the average value of the calendar life obtained from the test, which can be obtained from Equation (4); $k_\alpha$ is a one-sided tolerance coefficient to meet the reliability $\alpha$ with a given confidence level, which can be obtained by consulting the relevant standard [14], for the value under the normal distribution function, which can be approximated from Equation (5); and $S$ is the standard deviation for the calendar life obtained from the test, which can be obtained from Equation (6) [1–3,5,6]:

$$\overline{N} = \frac{1}{n} \cdot \sum_{i=1}^{n} N_i \tag{4}$$

$$k_\alpha = \frac{\mu_\alpha + \mu_\gamma \sqrt{\frac{1}{n}\left[1 - \frac{\mu_\gamma^2}{2(n-1)}\right] + \frac{\mu_\gamma^2}{2(n-1)}}}{1 - \frac{\mu_\gamma^2}{2(n-1)}} \tag{5}$$

$$S = \sqrt{\frac{1}{n-1} \cdot \sum_{i=1}^{n} \left(N_i - \overline{N}\right)^2} \tag{6}$$

In the above equations, $n$ is the number of data, $N_i$ is the calendar life of the $i$-th specimen, $\mu\alpha$ is the standard normal bias associated with the reliability $\alpha$, and $\mu\gamma$ is the standard normal bias associated with the confidence $\gamma$.

When the number of structures in the actual fleet is $m$, there is a confidence level of $\gamma$, whereby approximately $m(1 - \alpha)$ structures have failed in the protection system when the calendar safety life $N\alpha$ is reached.

Similarly, for an arbitrary reliability $\alpha'$, when the calendar safety life $N\alpha'$ determined by this reliability is reached, there is a confidence level of $\gamma$ that the protection system for approximately $m(1 - \alpha')$ structures fail.

(3)    Establish a model of the relationship between the maintenance cost and reliability, and find the reliability value that minimizes the maintenance cost of the fleet

Let the cost for the economic repair of the structure be $C_0$, the cost for replacing the structure be $C_1$, the total cost for the repair of the entire fleet of $m$ structures be $C$, and other costs for the fleet repair (such as disassembly, transportation, etc.) be $C_2$.

The damage threshold for a structure to reach the replacement requirement is $h_C$, and the average corrosion damage to the structural substrate versus the corrosion time is modeled as $h = f(t)$. This means that approximately half of the structural substrates with failed protection systems require replacement after a time of $t_C = f - 1(h_C)$. Assuming a reliability of $\alpha'$ ($\alpha' > \alpha$), among the $m(1 - \alpha)$ pieces of protection system failure structures that reach a calendar safety life of $N\alpha$, there are m$(1 - \alpha')$ pieces of structural substrates that experience at least $t_C$ time in a corrosive environment, i.e., $N\alpha - N\alpha' = t_C$, where $N\alpha'$ is the calendar safety life determined by the reliability $\alpha'$. At a calendar safety life of $N\alpha$, the

number of structural pieces that reach the damage threshold required for replacement is $0.5 \times m(1 - \alpha')$. Therefore, the following equation can be obtained [1,3,6]:

$$N_\alpha - N_{\alpha'} = \overline{N} - k_\alpha \cdot S - (\overline{N} - k_{\alpha'} \cdot S) = (k_{\alpha'} - k_\alpha) \cdot S = f^{-1}(h_C) \tag{7}$$

That is,

$$k_{\alpha'} = \frac{f^{-1}(h_C)}{S} + k_\alpha \tag{8}$$

The relationship between the reliability $\alpha$ and $\alpha'$ can be expressed by the following equation [1,3,6]:

$$\alpha' = g(\alpha) \tag{9}$$

Therefore, when a reliability of $\alpha$ is selected to determine the calendar safety life of the protection system, the total fleet maintenance cost to reach the calendar safety life of the protection system is [1–3]:

$$C = C_0 \cdot \{m(1 - \alpha) - 0.5m[1 - g(\alpha)]\} + C_1 \cdot 0.5m[1 - g(\alpha)] + C_2 \tag{10}$$

The total maintenance cost for the fleet at a single overhaul is not yet a measure of the economy of overhauling the fleet over its entire life. If the selected protection system calendar safety life reliability is very high, it makes the fleet's single overhaul costs very low, but increases the number of repairs. In terms of all the overhauls in the whole life of the fleet, the sum of the cost may be very high. Therefore, the overhaul economy should be measured by the overhaul cost $C_\alpha$ apportioned per year of service of the fleet [1,3]:

$$C_\alpha = \frac{C}{N_\alpha} = \frac{C}{\overline{N} - k_\alpha \cdot S} \tag{11}$$

In Equations (10) and (11), $m$, $C_0$, $C_1$, $C_2$, $\overline{N}$, and $S$ are known quantities. Therefore, the overhaul cost for the fleet is a function of the reliability $\alpha$ of the calendar safety life of the protection system. By taking the derivative of Equation (11) to 0, the value of the reliability $\alpha$ that minimizes the maintenance cost of the fleet is obtained.

(4)　Determining the reliability constraints

The reliability $\alpha$ that minimizes the maintenance cost of the fleet can be determined by the above steps, but it is not yet certain that this reliability is the calendar safety life reliability to be selected for the protection system. From the point of view of structural safety, the selected calendar safety life reliability for the protection system must also ensure that the aircraft structure has high reliability without failure fractures. That is, the reliability of the calendar safety life of the protection system is limited by the safety of the structure.

The damage threshold for a structure to reach scrapping requirements is $h_B$. That is, approximately half of the structural matrix for a protection system failure requires scrapping after a time of $tB = f - 1(h_B)$. Assuming a reliability of $\beta$, among the $m(1 - \alpha)$ pieces in protective system failure structures that reach the calendar safety life $N\alpha$, there are $m(1 - \beta)$ structural substrates that experience at least $t\beta$ time in the corrosive environment. Then, $N_\alpha - N_\beta = t\beta$, where $N\beta$ is the calendar safety life determined by the reliability $\beta$; at the calendar safety life $N\alpha$, the number of structural pieces that reach the critical damage value for the scrapping requirement is $0.5 \times m(1 - \beta)$. Therefore, the following equation can be obtained [1,3,6]:

$$N_\alpha - N_\beta = \overline{N} - k_\alpha \cdot S - (\overline{N} - k_\beta \cdot S) = (k_\beta - k_\alpha) \cdot S = f^{-1}(h_B) \tag{12}$$

That is,

$$k_\beta = \frac{f^{-1}(h_B)}{S} + k_\alpha \tag{13}$$

The functional relationship between the reliability $\alpha$ and $\beta$ is as follows [1,3,6]:

$$\beta = l(\alpha) \tag{14}$$

Then, the reliability of the structural substrate is [1,3,6]:

$$\lambda = 1 - \frac{0.5m(1-\beta)}{m} = 0.5[1 + l(\alpha)] \tag{15}$$

If the actual reliability of the structural substrate determined by the value of $\alpha$ meets the structural reliability requirements, the value of $\alpha$ can be selected as the reliability of the protection system. If not, the minimum reliability requirement of the structural substrate $\underline{\lambda}$ (e.g., 99.9%) needs to be inferred from Equation (15), to determine the minimum required value for the calendar life reliability of the protection system [1,3,6]. That is,

$$\underline{\alpha} = l^{-1}(2\underline{\lambda} - 1) \tag{16}$$

## 4. Example Analysis

### 4.1. Protection System Failure Test

A schematic diagram of the connection parts in the aircraft structure is shown in the left half of Figure 3. The structural connection form is a single-sided rivet lap type. The upper and lower wall panels are connected by three rows of semi-circular head rivets, and the number of rivets used for the connection between the two adjacent partition frames is 54 × 3. The thickness of the wall plate is 2 mm, and the material is 7B04 aluminum alloy. The rivet diameter is 5 mm, and the material is 2024-T4 aluminum alloy. The load direction for the structure is vertical in Figure 3. According to the characteristics of the hypothetical structure, the designed structural simulation specimen adopts the form of a multi-rivet lap, and the specimen is a single-sided lap riveted by semi-circular head rivets. The lap plate is attached according to the rolling direction. The protective layer on the surface of the specimen is epoxy primer. The configuration dimensions for the specimen are shown in the right half of Figure 3.

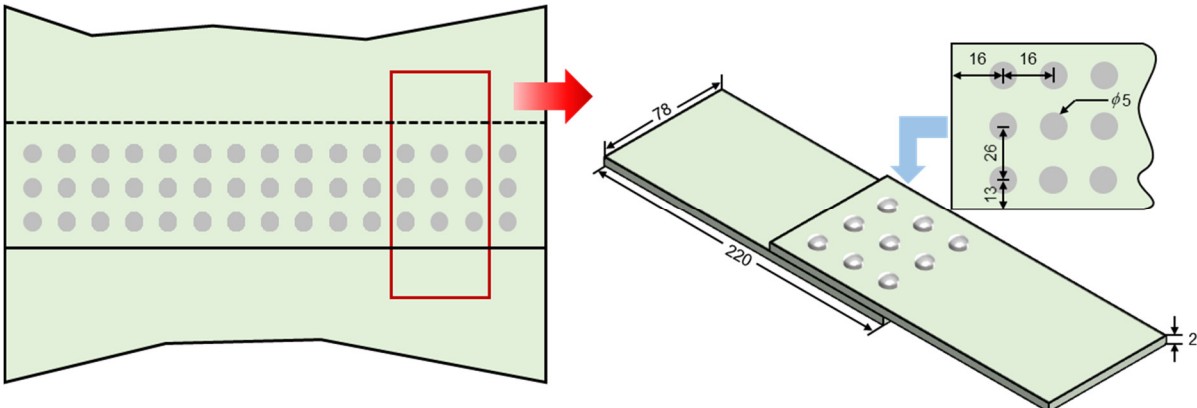

**Figure 3.** The specimen with the material and specific size.

The type of load spectrum used in this paper is the program block spectrum, the specific values are shown in Table 1, and the specific loading history is shown in Figure 4, with one spectrum block for every 1239 cycles [32]. The fatigue test loading equipment is an MTS-810-500 kN fatigue testing machine, the loading frequency is 15 Hz, and the test loading dynamic load error is less than 1% of the maximum load [32].

**Table 1.** Program block spectrum.

| Number of Program Blocks | 1 | 2 | 3 | 4 | 5 | 6 | 7 | 8 | 9 | 10 | 11 |
|---|---|---|---|---|---|---|---|---|---|---|---|
| $F_{max}$/kN | 13.88 | 18.50 | 22.63 | 22.63 | 24.75 | 22.63 | 22.63 | 20.63 | 26.75 | 16.50 | 15.25 |
| $F_{min}$/kN | 2.06 | 4.13 | 18.50 | 2.06 | 0.00 | 4.13 | 10.31 | 4.13 | 6.19 | 0.00 | 4.13 |
| Cycles | 305 | 28 | 12 | 31 | 3 | 10 | 202 | 95 | 3 | 123 | 427 |

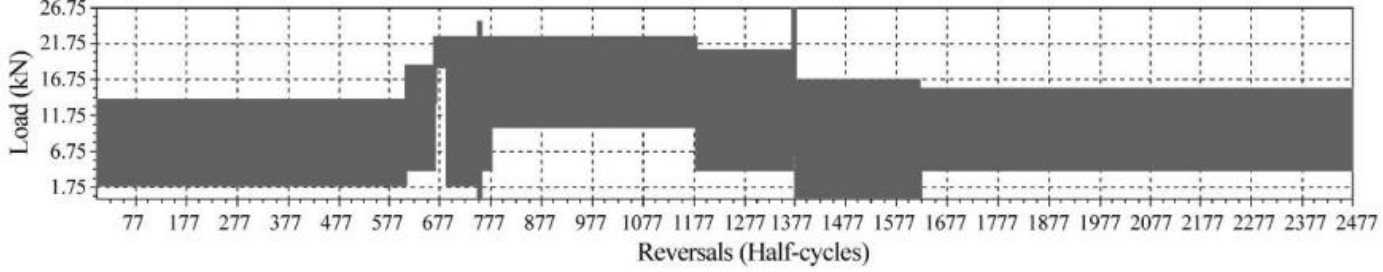

**Figure 4.** Fatigue test loading history.

Alternating tests on the specimens for corrosion and fatigue were carried out. Four specimens were subjected to a pure fatigue test. The corrosion fatigue alternating test was divided into five groups. The total corrosion time was 325 h, 975 h, 1300 h, 1625 h, and 1950 h, and there were four specimens in each group. The test sequence for the five groups in the corrosion fatigue alternating test is in the form of "corrosion–fatigue–corrosion–fatigue", that is, a twice alternating corrosion fatigue test. The corrosion time per corrosion and the number of cycles in the first fatigue loading are shown in Table 2. The specimen is loaded directly onto the fracture during the second fatigue load. Table 3 shows the failure time of the protection system for 16 specimens. Figure 5 shows the microscopic morphology for the failed part of the coating. As can be seen from the figure, the coating failure has produced many cracks and pits on the surface, allowing corrosive media to penetrate through the surface defects. From the sectional point of view, the coating has been spalled and the metal has been eroded by the corrosive medium, and severe intergranular corrosion (IGC) has occurred.

**Table 2.** Conditions for applying the corrosion fatigue alternating test.

| Group | Corrosion Time (Hours) | Number of First Fatigue Loading Cycles |
|---|---|---|
| 1 | 162.5 | 80,000 |
| 2 | 487.5 | 60,000 |
| 3 | 650.0 | 50,000 |
| 4 | 812.5 | 50,000 |
| 5 | 975.0 | 45,000 |

**Table 3.** Failure time of the specimen protection system.

| Number of Alternating Times | 8 | 9 | 10 | 11 | 12 | 13 | 14 |
|---|---|---|---|---|---|---|---|
| Number of specimens where the protection system failed | 1 | 2 | 4 | 4 | 3 | 1 | 1 |

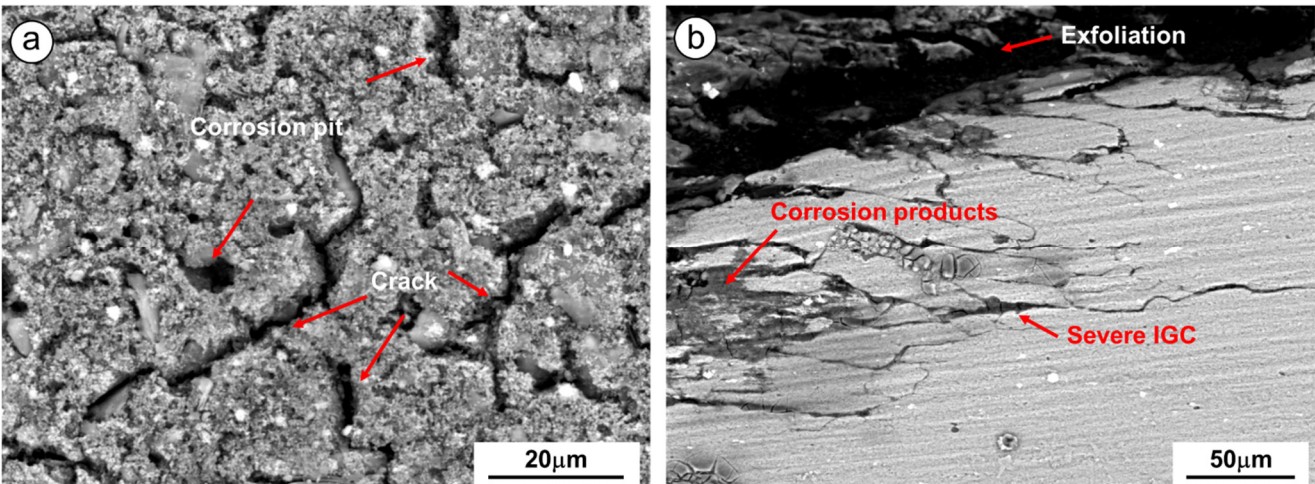

**Figure 5.** Microscopic morphology of the coating after failure: (**a**) surface morphology; (**b**) sectional morphology.

The failure of the protection system in the aircraft structure is the failure of the protection system near the rivet. The degradation of the fatigue life of the structure is such that the corrosion medium penetrates into the hole edge area along the rivet gap. During the overhaul, the initial protection state of the structure can basically be restored by replacing the rivets, repairing the hole edge, and repainting the protective layer, etc. As such, the test results can be used to analyze the calendar safety life before the first turn and the calendar safety life after the first turn of the protection system at the same time. That is, the calendar safety life before the first turn of the protection system and the calendar safety life after the first turn are equal. The pure fatigue test results for the specimen are shown in Figure 6. The results of the corrosion fatigue alternation test are shown in Figure 7.

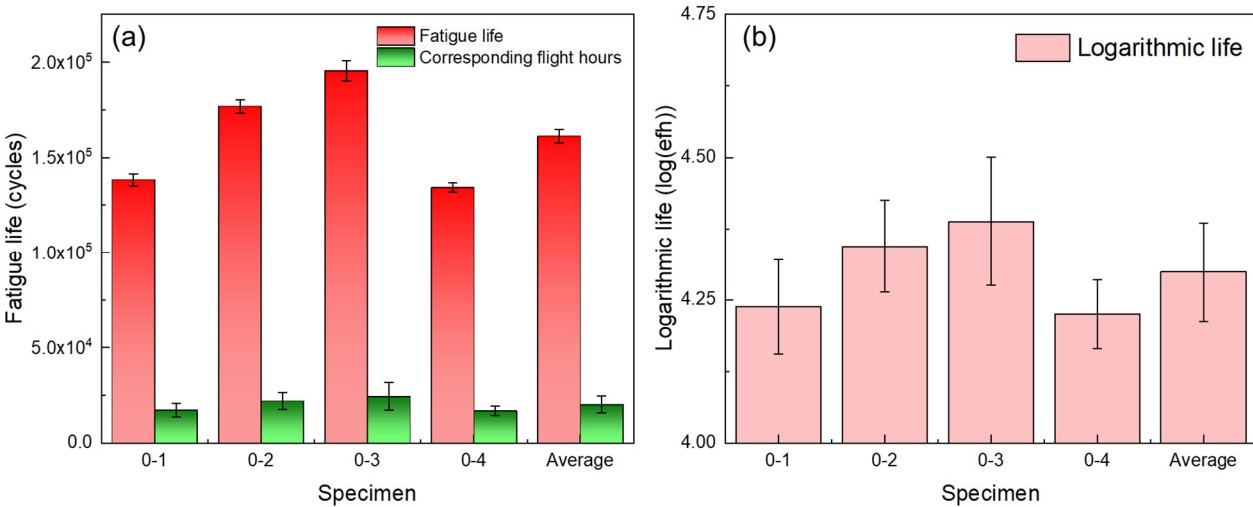

**Figure 6.** Results of the fatigue test: (**a**) fatigue life and corresponding flight hours; (**b**) logarithmic life.

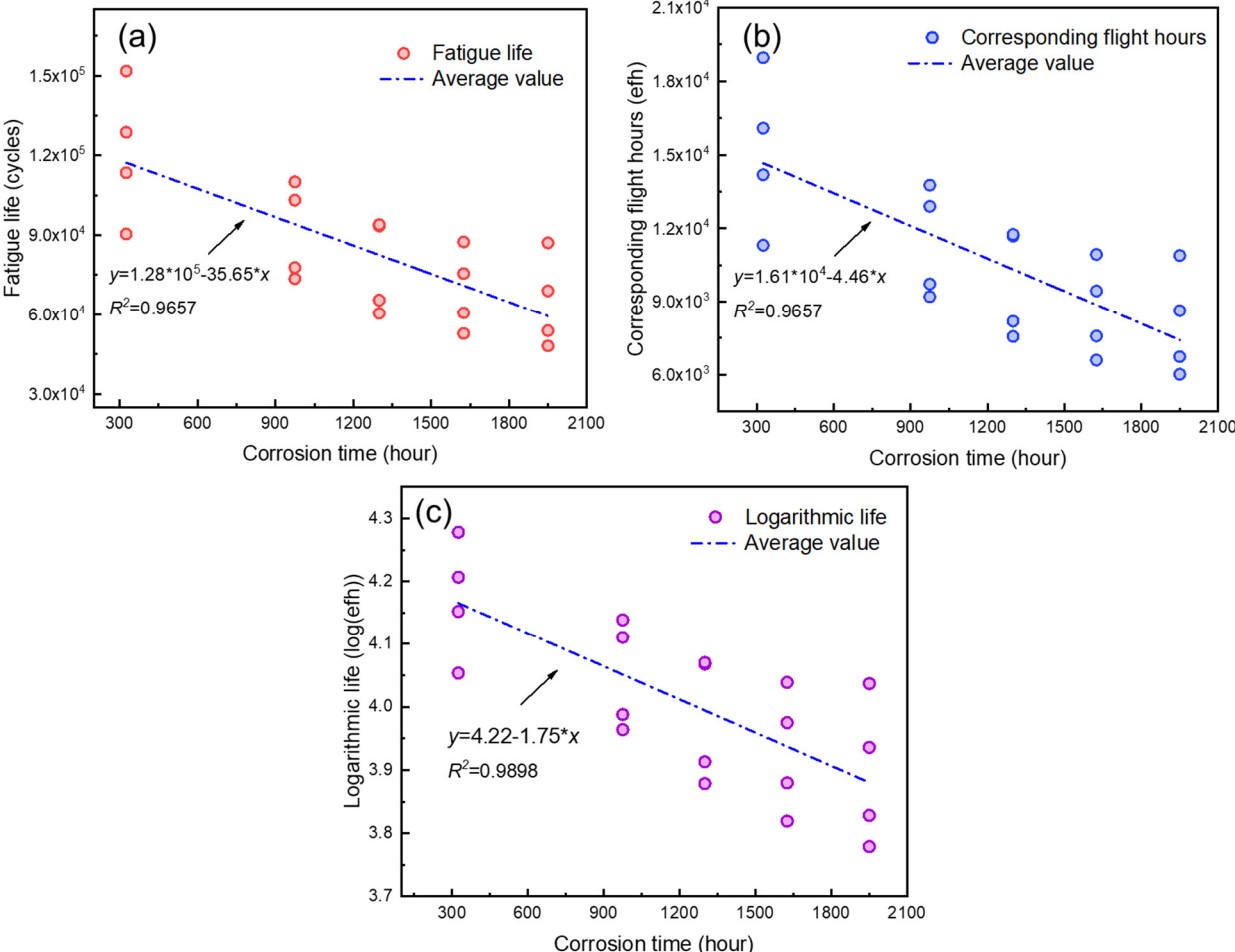

**Figure 7.** Results of the corrosion fatigue alternation test: (**a**) fatigue life; (**b**) corresponding flight hours; (**c**) logarithmic life.

*4.2. Determining the Reliability of the Safety Life of the Protection System Calendar*

The method proposed above is used to determine the reliability of the calendar safety life of the protection system in the research object. The reliability of the calendar safety life reflects the degree of safety of the protection system itself and has nothing to do with where the aircraft is in service. Thus, in order to simplify the calculation and unify the time measurement scale for an accelerated corrosion spectrum and an aging coating spectrum, the equivalent number of corrosion years of structural service in A is used as the time unit.

(1)  Establish a model of the relationship between the amount of corrosion damage with the structural matrix and the corrosion time

The amount of matrix corrosion damage can be directly expressed by parameters, such as the corrosion depth, corrosion weight loss, resistance change, etc., or it can be indirectly reflected by the degradation in the life of the structure. That is, it is considered that the degradation in the life of the structure after a certain period of corrosion reaches the critical value for corrosion damage.

Therefore, the corrosion influence coefficient curve can be used as the relationship between the amount of corrosion damage and the corrosion time in the structural matrix [12,15]. The general logarithmic form of the corrosion effect coefficient is as follows [11]:

$$\lg[1 - h] = b\lg T + \lg a \tag{17}$$

where $h$ is the average corrosion damage in the matrix indirectly reflected by the corrosion influence coefficient, $T$ is the equivalent corrosion years, and $a$ and $b$ are the fitting parameters.

According to the test corrosion time, as well as the test results in Tables 2 and 3, the data required for the corrosion effect coefficient curve fitting are listed in Table 4. Here, $h_i$ is the ratio of the average structure life with accelerated corrosion for $t$ hours in the laboratory to the average life of the pure fatigue test, that is, the corrosion influence coefficient of 50% of the data reliability, $h_i = N_{50}(T)/N_{50}(0)$.

**Table 4.** Data used for the corrosion impact factor curve fitting.

| Equivalent Corrosion Years $T$ of a-Site (a) | 0 | 10 | 30 | 40 | 50 | 60 |
|---|---|---|---|---|---|---|
| Mean life $N_{50}(t)$ (efh) | 20,147.2 | 15,136.0 | 11,394.6 | 9794.3 | 8636.3 | 8064.4 |
| Corrosion impact factor $h_i$ | 1.0000 | 0.7513 | 0.5656 | 0.4861 | 0.4287 | 0.4003 |

From Equation (17), least squares fitting is used to derive the amount of structural matrix corrosion damage and the corrosion time relationship model, as follows:

$$h = f(T) = 1 - 0.0784T^{0.5041} \tag{18}$$

The correlation coefficient is $R^2 = 0.997$. According to the requirements for the structural overhaul, when the corrosion impact factor is reduced to 0.8, the structure reaches the replacement damage threshold value $h_C$. That is, structural repair is no longer economical, or the repaired structure cannot guarantee the safety of flight and structural replacement is needed [10]. According to the requirements on the residual strength of the structure, when the corrosion impact factor is reduced to 0.6, the structure reaches the scrap damage threshold value $h_B$. That is, the structure can no longer continue to be used, owing to the risk of a sudden fracture [11].

(2)　Establish the calendar safety life expression for the protection system

According to the data in Table 2, the mean value for the calendar life of the protection system is 10.81 and the standard deviation is 1.56, calculated from Equations (4) and (6), respectively. Assuming that the calendar life of the protection system follows a normal distribution, the expression for the calendar life of the protection system, according to Equations (3) and (5) is:

$$N_\alpha = 10.81 - 1.56 \times \frac{\mu_\alpha + \mu_\gamma \sqrt{\frac{1}{16}\left[1 - \frac{\mu_\gamma^2}{30}\right] + \frac{\mu_\alpha^2}{30}}}{1 - \frac{\mu_\gamma^2}{30}} \tag{19}$$

where $\mu_\alpha$ is the standard normal bias associated with the reliability $\alpha$, and $\mu_\gamma$ is the standard normal bias associated with confidence $\gamma$.

Assuming that the number of fuselage wall panel attachment structures in the actual fleet is 1000, there is $\gamma$ confidence that at most $1000(1 - \alpha)$ structures will fail in the protection system when the calendar safety life $N\alpha$ corresponding to an arbitrary reliability $\alpha$ is reached.

(3)　Establish the relationship between the maintenance costs and the reliability model, and find the reliability value that minimizes the maintenance costs for the fleet

Assume that repair cost for the fuselage wall plate connecting structure is $C_0 = 20,000$ CNY/piece, that the replacement cost is $C_1 = 300,000$ CNY/piece, and that the total costs for other repairs to the 1000 pieces in the fuselage wall plate connecting structure in the fleet (including transportation and disassembly costs) is $C_2 = $ CNY 30 million.

According to Equations (10) and (11), the annual apportioned overhaul costs for the cluster when the calendar safety life of the protection system is reached are:

$$C_\alpha = \frac{19,000 - 14,000 g(\alpha) - 2000\alpha}{N_\alpha} \tag{20}$$

where $g(\alpha)$ is the expression of a particular reliability value $\alpha'$. According to Equations (8) and (18), the relationship between the reliability $\alpha$ and $\alpha'$ of the sought protection system calendar safety life is:

$$k_{\alpha'} = k_\alpha + 4.11 \tag{21}$$

The expression for $k_\alpha$ is shown in Equation (5).

Therefore, the expression for $g(\alpha)$ can be obtained according to Equation (21), which leads to the expression for $C_\alpha$. Then, the expression for $C_\alpha$ can be derived to obtain the value of the reliability $\alpha$ of the protection system that minimizes the overhaul costs of the fleet. With a confidence level of 0.90, the minimum value of $C_a$, the annual apportioned maintenance cost of the fleet before overhaul, is 3,876,600 CNY/year, corresponding to a reliability $\alpha$ of 0.763.

(4)     Reliability verification

According to Equations (13)–(15), the calendar safety life reliability for the protection system $\alpha$ = 0.763 corresponds to a structural reliability of 1. That is, from the perspective of the calendar safety life, it is basically impossible for the structure to fail before reaching overhaul. Through the above analysis, the calendar safety life reliability for the protection system in this model of the fuselage wall plate connection structure is 0.763. When the calendar safety life reliability of the protection system is taken as 0.999 according to the existing fatigue life reliability, the corresponding calendar safety life for the protection system is 4.41 years, and the annual maintenance cost $C_a$ shared by the fleet is 6.8073 million CNY/year. This not only increases the cost of maintenance, but also affects the combat readiness of the aircraft.

### 4.3. Determining the Calendar Safety Life of the Protection System

According to the protection system aging test basic spectrum block for A, B, C, and D, four regions of equivalent corrosion years for conversion, and the data in Figure 7, the data used to determine the calendar safety life of the protection system in the four regions are listed in Table 5.

**Table 5.** Equivalent failure time of the protection system under four regions, A, B, C, and D.

| Number of Specimens with Protection System Failure | 1 | 2 | 4 | 4 | 3 | 1 | 1 |
|---|---|---|---|---|---|---|---|
| Corrosion (aging) basic block | 8 | 9 | 10 | 11 | 12 | 13 | 14 |
| Equivalent corrosion years in A | 8.00 | 9.00 | 10.00 | 11.00 | 12.00 | 13.00 | 14.00 |
| Equivalent corrosion years in B | 5.33 | 6.00 | 6.67 | 7.33 | 8.00 | 8.67 | 9.33 |
| Equivalent corrosion years in C | 4.00 | 4.50 | 5.00 | 5.50 | 6.00 | 6.50 | 7.00 |
| Equivalent corrosion years in D | 3.20 | 3.60 | 4.00 | 4.40 | 4.80 | 5.20 | 5.60 |

According to the data in Table 1, the average values for the calendar life of the protection system in A, B, C, and D, can be calculated from Equations (4) and (6) as 10.81, 7.21, 5.41, and 4.33 years, respectively, with standard deviations of 1.56, 1.04, 0.78, and 0.62 years, respectively. Since the surface area of the specimen at the test part (rivet lap) is 1/18 of the actual structure, the reliability of the selected calendar safety life is 76.3%. According to Equation (1), the reliability $\alpha'$ that should be selected for the analysis of the test data obtained from the specimen is (18 + 0.763 − 1)/18 = 98.68%.

Using the calculation from Equation (5), the reliability 98.68% and confidence 90% corresponds to the normal distribution one-sided tolerance coefficient $k_\alpha = 2.9897$. According to Equation (3), the calendar safety life of the fuselage wall plate connection structure in four places, A, B, C, and D, with 76.3% reliability and 90% confidence is 6.14 years, 4.10 years, 3.08 years, and 2.48 years, respectively.

Since this paper presents a new principle and method, there are no results from other literature to compare. The study in this paper is about reliability, which can be found in the literature [7,33]. The methods used are based on the average value, which means that the reliability is chosen to be 50%. The method in this paper finally determines the reliability to be around 70%, which can achieve both the safety requirements and makes the process more economical.

## 5. Conclusions

In this paper, the principles and methods for determining the calendar safety life of aircraft structural protection systems were studied from the perspective of reliability.

(1) The methods and specific technical approaches to determine the calendar safety life for two types of aircraft structural protection systems in a single service environment and in multiple service environments were established. The method for determining the calendar safety life of the protection system can lay the foundation for determining the overall calendar safety life of the structure;

(2) The method for determining the reliability of the calendar safety life of the aircraft structure protection system was studied. Based on the relationship between the corrosion damage of the structural substrate and the corrosion time and the expression of the calendar safety life of the protection system, the expression for the relationship between the maintenance cost of the fleet and the reliability of the calendar safety life of the aircraft structure protection system was established. The reliability value for the lowest maintenance cost of the fleet can be obtained by taking 0 for the derivative of the formula. The reliability limit is established from the perspective of structural safety in service. The reliability of the calendar safety life of a reasonably selected protection system is important to ensure the safety of the aircraft structures, improve the efficiency of use, and reduce the maintenance costs;

(3) Taking a hypothetical aircraft fuselage wall plate connection structure as an example, an alternating corrosion fatigue test using protection system specimens was carried out. Based on the test results, an example for determining the life of a structural protection system under hypothetical service conditions and service environments was given. This included determining the calendar safety life reliability of the protection system and determining the calendar safety life of the protection system. In general, the reliability of the calendar safe life of a structure is 99.9%, and after the analysis in this paper, the reliability of the structural protection system is about 70%.

**Author Contributions:** Conceptualization, T.Z. (Tianyu Zhang) and T.Z. (Teng Zhang); validation, Y.H., T.Z. (Tianyu Zhang) and T.Z. (Teng Zhang); formal analysis, T.Z. (Tianyu Zhang); investigation, B.H. and C.L.; writing—original draft preparation, T.Z. (Tianyu Zhang); writing—review and editing, T.Z. (Teng Zhang) and B.H.; supervision, Y.H.; funding acquisition, Y.H. All authors have read and agreed to the published version of the manuscript.

**Funding:** The authors gratefully acknowledge the financial support from the National Science and Technology Major Special Funding (No.: J2019-I-0016-0015), the National Natural Science Foundation of China (52175155) and the National Natural Science Foundation of China (52005507).

**Institutional Review Board Statement:** Not applicable.

**Informed Consent Statement:** Not applicable.

**Data Availability Statement:** All data included in this study are available upon request by contacting the corresponding author.

**Conflicts of Interest:** The authors declare no conflict of interest.

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
