# Peer review of "Principle and Method for Determining the Calendar Safety Life of Aircraft Structural Protection Systems"

_coatings, doi:10.3390/coatings13060976_

Round 1

Reviewer 1 Report

This manuscript is about the principle and method of determining the calendar safety life of aircraft structural protection systems. However, there is no structure for the scientific article. It seems to be a chapter book or a course project! This technical issue could also be found from the low number of references. Moreover, insufficient data were presented and no discussion could be found on obtained results. In addition, all formulations need references and more examples must be presented.  

Author Response

Thanks to the reviewer for this suggestion. In fact, this paper differs from papers that focus on experimental studies or simulation studies in that it presents the principles and methods for determining the calendar safety life of aircraft structural protection systems. Therefore, the reviewers may be confused about the structure of this paper. The structure of the paper has been adjusted for the convenience of readers. The first part of the main text is “Principles and methods for determining the calendar safety life of aircraft structural protection systems in single and multiple service environments”, and contains “Principles and methods for determining the calendar safety life of aircraft structural protection systems in a single service environment” and "Principles and methods for determining the calendar safety life of aircraft structural protection systems in multiple service environments". The second part of the main text is “Principles and methods for determining the calendar safety life reliability of aircraft structural protection systems”. The third part of the main text is “Example analysis”, and contains “Protection system failure test”, “Determining the reliability of the safety life of the protection system calendar” and “Determining the calendar safety life of the protection system”.

We are very sorry that we did not cite all the references for the models and computational formulas before, and we have now revised the references in the full text and added references that were not cited before.

In fact, we perform an experimental validation of the method proposed in this paper in the fourth part of the paper, carrying out tests on typical connection structures of aircraft with protection systems. Taking a hypothetical aircraft fuselage wall plate connection structure as an example, an alternating corrosion fatigue test with protection system specimens was carried out. Based on the test results, an example of determining the life of a structural protection system under hypothetical service conditions and service environments was given. This included determining the calendar safety life reliability of the protection system and determining the calendar safety life of the protection system.

Reviewer 2 Report

Dear author, 

It is impressive that you investigated the safety of aircraft structures, improve the efficiency of use, reduce maintenance costs and created this paper. I would also like to congratulate you on writing in a well-structured and understandable manner. However, it would be better if you add some more references to improve your manuscript.

Author Response

We thank the reviewer for the positive comments. We have added references as suggested by the reviewers.

Reviewer 3 Report

The article "Principle and method of determining the calendar safety life of aircraft structural protection systems" is devoted to the actual topic (safety of the surface protection system for aircraft structures). The authors described the methodology in detail. The results obtained are of particular interest to a wide range of readers, including those from related fields of research. The schemes are clear and understandable. The figures are of good quality. References are relevant and related to the research. In general, the article was submitted at a high level.

I recommend accepting the article in its present form.

Author Response

We appreciate the positive comments from the reviewer.

Reviewer 4 Report

The authors have worked on an interesting area. However the depth of research needs improvement before the article can be accepted a journal paper:

1. The authors focus on the surface protection system for aircrafts. The should discuss in detail the need for such protection and the materials and mechanisms involved in surface protection. What are the primary components of concern against corrosion protection and the materials used to fabricate them.

2. How does surface surface deterioration affect mechanical properties like fatigue, yield strength etc.

3. The research should identify load bearing and non load bearing parts and classify them in terms of materials and apply the research accordingly.

4. The chemical and mechanical properties of the alloys considered should be presented in tabular form including density, fatigue life, yield strength, tensile strength, hardness, toughness etc.

5. There are several language usage errors that should be corrected. Overall, the manuscript should be proofread by an English speaker/professor/instructor. A formal technical reporting style should be followed and first person reporting should be avoided.

There are several language usage errors that should be corrected. Overall, the manuscript should be proofread by an English speaker/professor/instructor. A formal technical reporting style should be followed and first person reporting should be avoided.

Author Response

Reviewer #4:

The authors have worked on an interesting area. However the depth of research needs improvement before the article can be accepted a journal paper.

  1. Response to comment:

The authors focus on the surface protection system for aircrafts. The should discuss in detail the need for such protection and the materials and mechanisms involved in surface protection. What are the primary components of concern against corrosion protection and the materials used to fabricate them.

Response: Thanks to the reviewer for this suggestion. Protective systems such as coatings applied on the surface of aircraft structures can effectively protect the structure from corrosion and ensure the safe flight of the aircraft [1]. Surface coating systems play a very important protective role for aircraft performing cruise missions in severe weather areas (e.g., high temperature, high humidity, high salt, high sand, strong UV, etc.) [2, 3]. The complete protective system for aircraft structures consists of an anodic oxide film, primer, intermediate and topcoat [2]. However, protective coatings are used in many cases as single or separate coating systems, which are related to the specific service environment and operating conditions. Moreover, the correct combination of the pre-treatment method of the substrate and the protective coating system determines the corrosion resistance of the whole protection system. Aircraft that are often in service in coastal areas, island areas or industrial areas are subject to harsh environmental corrosion [4-7]. Moreover, because the aircraft is a special closed structure, water often accumulates inside it and causes corrosion, such as the seam area between the fuselage and the wing skin, the connection area between the bolts or rivets and the plates, the landing gear of the aircraft, the area where the battery is placed, the recess area between the flaps and the hinges where water easily accumulates, the area affected by the engine exhaust, and other places where water easily accumulates [8].

We have added the following paragraph to the “Introduction” section:

“Protective systems such as coatings applied on the surface of aircraft structures can effectively protect the structure from corrosion and ensure the safe flight of the aircraft [1]. Surface coating systems play a very important protective role for aircraft performing cruise missions in severe weather areas (e.g., high temperature, high humidity, high salt, high sand, strong UV, etc.) [2, 3]. The complete protective system for aircraft structures consists of an anodic oxide film, primer, intermediate and topcoat [2]. However, protective coatings are used in many cases as single or separate coating systems, which are related to the specific service environment and operating conditions. Moreover, the correct combination of the pre-treatment method of the substrate and the protective coating system determines the corrosion resistance of the whole protection system. Aircraft that are often in service in coastal areas, island areas or industrial areas are subject to harsh environmental corrosion [4-7]. Moreover, because the aircraft is a special closed structure, water often accumulates inside it and causes corrosion, such as the seam area between the fuselage and the wing skin, the connection area between the bolts or rivets and the plates, the landing gear of the aircraft, the area where the battery is placed, the recess area between the flaps and the hinges where water easily accumulates, the area affected by the engine exhaust, and other places where water easily accumulates [8].”

  1. Response to comment:

How does surface deterioration affect mechanical properties like fatigue, yield strength etc.

Response: Thanks to the reviewer for this suggestion. The surface of the protection system will gradually undergo physical or chemical changes when exposed to strong ultraviolet radiation, high temperature, high humidity and high salt, resulting in a gradual decline in the protective properties of the coating, thus exposing the substrate to the erosion of corrosive media. The failure of the protective properties of the coating will seriously affect the structural integrity of the aircraft [9, 10]. Some research studies have pointed out the frequency of accidents due to corrosion problems or fatigue fracture problems resulting from corrosion, which account for approximately 20% of all accidents occurring in aircraft [11, 12]. For example, in 1981, the coating on the surface of an aircraft fuselage failed, resulting in corrosion and fatigue cracks due to corrosion problems, which eventually disintegrated in the air [12]. in 2002, the skin of a passenger aircraft was scuffed at the tail and the coating peeled off, which was not serviced in time, resulting in corrosion and fatigue cracks in the substrate, which eventually crashed into the sea, killing 225 people [13]. When the aircraft was in service on Hainan Island, China, the coating on the skin surface began to break down after six months, the aluminum alloy substrate began to corrode after one year, and the surface turned gray and lost its metallic luster after five years, with severe corrosion perforations occurring [2]. Aircraft in service in the plateau region generally do not consider the impact of corrosion, but the plateau region has a strong ultraviolet radiation, if the coating aging, and then perform the coastal area rotation mission is extremely prone to corrosion damage. As shown in Figure 1 is a type of aircraft corrosion caused by coating failure.

Figure 1. Corrosion of aircraft caused by coating failure.

  1. Response to comment:

The research should identify load bearing and non load bearing parts and classify them in terms of materials and apply the research accordingly.

Response: Thanks to the reviewer for this suggestion. Our research is focused on the force-bearing components of aircraft structures, and the main object of our research is the metallic structures and coating systems of aircraft surfaces. Based on the reviewer's comments, we have added the following paragraph to the “Introduction” section:

“Our research is focused on the force-bearing components of aircraft structures, and the main object of our research is the metallic structures and coating systems of aircraft surfaces.”

  1. Response to comment:

The chemical and mechanical properties of the alloys considered should be presented in tabular form including density, fatigue life, yield strength, tensile strength, hardness, toughness etc.

Response: Thanks to the reviewer for this suggestion. Because this paper studies the principles and methods for determining the calendar safety life of aircraft structural protection systems, it is not specific to a particular material. The mechanical properties and other parameters of different types of aluminum or titanium alloys are different, so they are not listed separately in this paper.

  1. Response to comment:

There are several language usage errors that should be corrected. Overall, the manuscript should be proofread by an English speaker/professor/instructor. A formal technical reporting style should be followed and first person reporting should be avoided.

Response: Thanks to the reviewer for this suggestion, we are sorry for our negligence. We have revised the overall grammar of the paper, and thank International Science Editing for editing this manuscript. If there are any other grammatical problems, please point them out directly.

References:

[1]           Li J G. Anti-corrosion surface engineering technology [M]. Beijing: Chemical Industry Press, 2003.

[2]           He D. Aviation Coating and Painting Technology [M]. Beijing: Chemical Industry Press, 2000.

[3]           Chen Y L. Corrosion Control and Strength Assessment of Naval Aircraft Structures [M]. Beijing: Defense Industry Press, 2009.

[4]           Alias M N, Brown R. Corrosion behavior of carbon fiber composites in the marine environment[J]. Corrosion Science, 1993, 35(1): 395-402.

[5]           An T, He Y, Feng Y. Experimental Study on Compression Behavior of Fiber-Reinforced Resin-Based Composite Stiffened Panels in Hygrothermal Environment[J]. Strength of Materials, 2020, 52(4): 655-661.

[6]           Bano H, Mahmood A, Khan M I, et al. Synergistic Corrosion Mitigation Appraisal of Coal Tar Epoxy Duplex Coating System by Spectroscopic and Microscopic Techniques[J]. Arabian Journal for Science and Engineering, 2014, 39(9): 6783-6791.

[7]           Cao C N. Natural Environmental Corrosion of Chinese Materials[M]. Beijing: Chemical Industry Press, 2005.

[8]           Yu M, Liu J H, Li S M. Aviation aluminum alloy corrosion protection and inspection methods [M]. Beijing: Science Press, 2017.

[9]           Defense U D O. Aircraft structural integrity program(ASIP) [S]. Department of defense standard MIL-STD-1530D(USFA). 2016.

[10]         Mu Z T, Li X D, Liu Z G. Environmental corrosion and fatigue analysis of aircraft structural materials [M]. Beijing: Defense Industry Press, 2014.

[11]         Zeng R C, Han E H. Corrosion and protection of materials [M]. Beijing: Chemical Industry Press, 2006.

[12]         Luo Y S, Cheng P Q. Experience and Lessons Learned - Aircraft Corrosion Damage Case Study [M]. Xi'an: Northwestern Polytechnic University Press, 2016.

[13]         Zhang T. Research on the principle and some key issues of aircraft structure life control under the influence of corrosive environment [D]. Xi'an: Air Force Engineering University, 2015.

Special thanks to you for your good comments.

We tried our best to improve the manuscript and made some changes in the manuscript. These changes will not influence the content and framework of the paper. And we marked in red in revised paper.

We appreciate for Editors/Reviewers’ warm work earnestly, and hope that the correction will meet with approval.

Once again, thank you very much for your comments and suggestions

Looking forward to hearing from you soon.

With kindest regards,

Yours Sincerely,

Zhang Tianyu.

Round 2

Reviewer 1 Report

As mentioned before, this manuscript has no structure for the scientific article. Now, very long titles are used for each part, which is unacceptable. Therefore, I must again reject it. Other comments can be found as follows,

1) No quantitative results could be seen in the abstract. 

2) Keywords must be found in the abstract or the title. 

3) No novelty could be found in the introduction, compared to the literature review.

4) All formulations need references. 

5) No repeatability of testing could be seen in this text. 

6) Fatigue data must be presented in figures, instead of tables. 

7) No discussion could be found on the obtained results. They must be compared to other results of other articles. 

8) "Conclusion" must be changed to "Conclusions". Moreover, no quantitative results could be seen again. 

9) SEM images must be provided for samples after failures. 

Author Response

Dear Editor and Reviewers,

Thank you for your letter and for the reviewers’ comments concerning our manuscript entitled “Principle and method of determining the calendar safety life of aircraft structural protection systems” (Manuscript Number: coatings-2353657). Those comments are all valuable and very helpful for revising and improving our paper, as well as the important guiding significance to our researches. We have studied comments carefully and have made correction which we hope meet with approval. Revised portion are marked in red in the paper. The main corrections in the paper and the responds to the reviewer’s comments are as flowing:

Responds to the reviewer’s comments:

Reviewer #1:

As mentioned before, this manuscript has no structure for the scientific article. Now, very long titles are used for each part, which is unacceptable. Therefore, I must again reject it. Other comments can be found as follows.

Response:

We are sorry that we have confused the reviewers. In general, the structure of a scientific and technical paper includes the following parts: experiment/model creation process, analysis of experiment/simulation results, and discussion. The contribution of this paper is to propose a principle and method that contains the following parts: principle and method of determining calendar safe life, principle and method of determining reliability, example analysis. The structure of the article was not problematic from the author's point of view. Based on the reviewer's suggestion, we shortened the word count of the title. The revised title is as follows:

“2. Principles and methods for determining the calendar safety life”

“2.1 Single service environment”

“2.2   Multiple service environments”

“3. Principles and methods for determining the calendar safety life reliability”

“4. Example analysis”

“4.1 Protection system failure test”

“4.2 Determining the reliability of the safety life of the protection system calendar”

“4.3 Determining the calendar safety life of the protection system”

  1. Response to comment:

No quantitative results could be seen in the abstract.

Response: Thanks to the reviewer for this suggestion. Based on the reviewers' comments, the following sentence was added to the abstract: “Generally speaking, the reliability of the calendar safe life of the structure is 99.9%, and after the analysis of this paper, the reliability of the structural protection system can be about 70%.”

  1. Response to comment:

Keywords must be found in the abstract or the title.

Response: Thanks to the reviewer for this suggestion. According to the reviewers' comments, we have revised the keywords as follows: “principle; method; protection system; calendar safety life; aircraft structure”.

  1. Response to comment:

No novelty could be found in the introduction, compared to the literature review.

Response: Thanks to the reviewer for this suggestion. The Introduction of this paper focuses on the definition, influencing factors, and classification of the calendar safety life of structural protection systems. We added the following paragraph at the end of the introduction: “This paper presents a principle and method for determining the calendar safe life of an aircraft structural protection system, proposes a method for determining its reliability, and finally validates and analyzes this method by means of an example.”

  1. Response to comment:

All formulations need references.

Response: Thanks to the reviewer for this suggestion. We apologize for our mistake. We have marked all formulas with citations.

  1. Response to comment:

No repeatability of testing could be seen in this text.

Response: Thanks to the reviewer for this suggestion. Our validation tests were divided into five groups with four specimens in each group, meeting the minimum number of test pieces required.

  1. Response to comment:

Fatigue data must be presented in figures, instead of tables.

Response: Thanks to the reviewer for this suggestion. Based on the reviewer's suggestion, we have replaced the original Tables 3 and 4 with figures. 

  1. Response to comment:

No discussion could be found on the obtained results. They must be compared to other results of other articles.  

Response: Thanks to the reviewer for this suggestion. In general, the reliability of the calendar safe life of a structure is 99.9%, and after the analysis in this paper, the reliability of the structural protection system can be about 70%. We have included the above statement in the conclusion.

  1. Response to comment:

"Conclusion" must be changed to "Conclusions". Moreover, no quantitative results could be seen again.

Response: Thanks to the reviewer for this suggestion. Based on the reviewer's suggestion, we have revised the “Conclusion” to “Conclusions” and added the following sentence at the end: “In general, the reliability of the calendar safe life of a structure is 99.9%, and after the analysis in this paper, the reliability of the structural protection system can be about 70%.”

  1. Response to comment:

SEM images must be provided for samples after failures.

Response: Thanks to the reviewer for this suggestion. Based on the reviewer's suggestion, we added SEM images in section 4.1 to illustrate the microscopic morphology of the coating after failure. And we added the following paragraph: "As can be seen from the figure, the coating failure has produced a large number of cracks and pits on the surface, allowing corrosive media to penetrate through the surface defects. From the sectional view, the coating has been spalled and the metal has been eroded by the corrosive medium and severe intergranular corrosion (IGC) has occurred."

Special thanks to you for your good comments.

We tried our best to improve the manuscript and made some changes in the manuscript. These changes will not influence the content and framework of the paper. And we marked in red in revised paper.

We appreciate for Editors/Reviewers’ warm work earnestly, and hope that the correction will meet with approval.

Once again, thank you very much for your comments and suggestions

Looking forward to hearing from you soon.

With kindest regards,

Yours Sincerely,

Zhang Tianyu.

Reviewer 4 Report

While all other observations have been addressed, the response to observations 3 and 4 should be revisited and addressed. The authors mention that their "research is focused on the force-bearing components of aircraft structures, and the main object of our research is the metallic structures and coating systems of aircraft surfaces.". In this context, some important/representative load bearing parts that the authors have in mind should be specified and listed along with the materials that are used to manufacture them. Also the authors mention that "this paper studies the principles and methods for determining the calendar safety life of aircraft structural protection systems, it is not specific to a particular material.". This is all the more reason to specify prospective materials and their properties for the interest of future research. Moreover there are still first person statements (I, we etc.) that should be avoided in a technical article.

The language usage is at times informal. First person statements should be avoided.

Author Response

Dear Editor and Reviewers,

Thank you for your letter and for the reviewers’ comments concerning our manuscript entitled “Principle and method of determining the calendar safety life of aircraft structural protection systems” (Manuscript Number: coatings-2353657). Those comments are all valuable and very helpful for revising and improving our paper, as well as the important guiding significance to our research. We have studied the comments carefully and have made corrections which we hope meet with approval. Revised portions are marked in red on the paper. The main corrections in the paper and the response to the reviewer’s comments are as flowing:

Responds to the reviewer’s comments:

Reviewer #4:

While all other observations have been addressed, the response to observations 3 and 4 should be revisited and addressed. The authors mention that their "research is focused on the force-bearing components of aircraft structures, and the main object of our research is the metallic structures and coating systems of aircraft surfaces.". In this context, some important/representative load bearing parts that the authors have in mind should be specified and listed along with the materials that are used to manufacture them. Also the authors mention that "this paper studies the principles and methods for determining the calendar safety life of aircraft structural protection systems, it is not specific to a particular material.". This is all the more reason to specify prospective materials and their properties for the interest of future research. Moreover there are still first person statements (I, we etc.) that should be avoided in a technical article.

Response: Thanks to the reviewer for this suggestion, we are sorry for our negligence. Based on the reviewer’s suggestion, we added the following sentence to illustrate the structural material in the Introduction: “The main substrate materials studied are metallic materials such as aluminum alloys, and the main coating materials studied are organic coatings such as epoxy primers.”

And we have revised the first-person narrative passages in the article.. If there are any other grammatical problems, please point them out directly.

Special thanks to you for your good comments.

We tried our best to improve the manuscript and made some changes in the manuscript. These changes will not influence the content and framework of the paper. And we marked in red in revised paper.

We appreciate for Editors/Reviewers’ warm work earnestly, and hope that the correction will meet with approval.

Once again, thank you very much for your comments and suggestions

Looking forward to hearing from you soon.

With kindest regards,

Yours Sincerely,

Zhang Tianyu.

Round 3

Reviewer 1 Report

1) Still, the structure is not scientific! The introduction, research method, results and discussion, conclusions, and references must be provided in the text.

2) The novelty must be highlighted in the introduction, compared to the literature review. This job was not done! A sentence at the end about the manuscript is not the highlight of the novelty.

3) Figure 4 needs the standard deviation to show the repeatability of testing.

4) The R2 value must be provided in Figure 5. What was the stress? What was the detail of testing? Which type of loading? 

5) The discussion is not extended! The obtained results must be, must be compared to the results of other articles. 

6)  The SEM images must be provided for the sample after the failure of fatigue. What is fracture behavior? Ductile or brittle? What were the features? Dimple? CLeavage? What was the failure mechanism? The EDX analysis must be provided. 

Author Response

Dear  Reviewer,

Thank you for your letter and for the comments concerning our manuscript entitled “Principle and method of determining the calendar safety life of aircraft structural protection systems” (Manuscript Number: coatings-2353657). Many thanks to you for your repeated comments and suggestions, which are greatly appreciated. Those comments are all valuable and very helpful for revising and improving our paper, as well as the important guiding significance to our research. We have studied the comments carefully and have made corrections which we hope meet with approval. Revised portions are marked in red on the paper. The main corrections in the paper and the response to the comments are as flowing:

  1. Response to comment:

Still, the structure is not scientific! The introduction, research method, results and discussion, conclusions, and references must be provided in the text.

Response: Thanks to the reviewer for this suggestion. As we have explained several times in our previous response letters, this paper is not different from scientific and technical papers that focus on experimental/model studies; the focus of this paper is to propose a principle and method to determine the calendar safety life. The following articles with a similar structure to this paper are available for reviewers to refer to:

[1] He Y T. On the generalized survivability and evaluation method of combat aircraft [J]. Acta Aeronautica et Astronautica Sinica, 2022, 42(in Chinese). doi: 10.7527/S1000-6893.2022.26118.

[2] HeY T. Study on Characterization Method of Aircraft Health Status [J]. Advances in Aeronautical Science and Engineering, 2021, 12(03): 1-8.

[3] He Y T. Aircraft recoverability and its design methods [J]. Journal of Air Force Engineering University (Natural Science Edition), 2019, 20(06): 1-8.

[4] He Y, Li C, Zhang T, et al. Service fatigue life and service calendar life limits of aircraft structure: aircraft structural life envelope[J]. Aeronautical Journal, 2016, 120(1233): 1746-1762.

[5] Li S A, Zhang H X, Li S L, et al. Research on Aircraft Survivability Evaluation and Synthetic Tradeoff

Methods [J]. Acta Aeronautica et Astronautica Sinica, 2005 ,26(1): 23-26.

[6] Li S A, SONG B F, LI D X, et al. A method for technology evaluation and selection of aircraft survivability design[J]. Electronics Optics & Control, 2008 ,15(7): 26-29.

[7] Cui R H, He Y T. Vulnerability Assessment of Military Aircraft Design[J]. Journal of Aircraft Design,

2007, 27(3): 56-58.

[8] Pei Y, Song B F, Li Z K. Research on the Aircraft Vulnerability Assessment Method[J]. Journal of Projectiles, Rockets, Missiles and Guidance, 2004, 24 (02): 70-74.

[9] Li J, Wu Y Z, Li B S, et al. Estimate Method for Calendar Life of Helicopter′s Coating Based on Electrochemical Impedance [J].Equipment Environmental Engineering, 2017, (7).

[10] Wood K. A natural-weathering approach to predict the color service life of fluoropolymer-based coatings with organic pigments[M]. Service Life Prediction of Polymers and Coatings, 2020.

[11] Shreepathi S, Guin A K, Naik S M, et al. Service life prediction of organic coatings: electrochemical impedance spectroscopy vs actual service life[J]. Journal of Coatings Technology & Research, 2011, 8(2): 191-200.

  1. Response to comment:

The novelty must be highlighted in the introduction, compared to the literature review. This job was not done! A sentence at the end about the manuscript is not the highlight of the novelty.

Response: Thanks to the reviewer for this suggestion. According to the reviewers' comments, we added the following paragraph to the preamble: “Failure of a protective coating on an aircraft structure does not necessarily mean that the structure itself has failed. The structure will be selected for a certain reliability in the life determination, and the current common practice is to select a reliability level of 50%. This means that the maintenance cost of the aircraft is too high. The research in this paper found that the reliability of the aircraft structural protection system can meet the requirements at about 70%, which means that this can greatly reduce maintenance costs and be more economical while meeting safety requirements.”

  1. Response to comment:

Figure 4 needs the standard deviation to show the repeatability of testing.

Response: Thanks to the reviewer for this suggestion. Based on the reviewers' comments, we have added standard deviations to the images to illustrate the reproducibility of the tests.

Please see the attachment for the figures.

Figure 6. Results of fatigue test: (a) fatigue life and corresponding flight hours; (b) logarithmic life.

  1. Response to comment:

The R2 value must be provided in Figure 5. What was the stress? What was the detail of testing? Which type of loading?

Response: Thanks to the reviewer for this suggestion. Based on the reviewer's comments, we added the fitting equation and R2 to the figure.

Please see the attachment for the figures.

Figure 7. Results of corrosion fatigue alternation test: (a) fatigue life; (b) corresponding flight hours; (c) logarithmic life.

The type of load spectrum used in this paper is the program block spectrum, the specific values are shown in Table 2, and the specific loading history is shown in Figure 4, with 1 spectrum block for every 1239 cycles. The fatigue test loading equipment is MTS-810-500kN fatigue testing machine, the loading frequency is 15Hz, the test loading dynamic load error are less than 1% of the maximum load. Specific references can be found in the literature (Zhang Teng. Research on the principle and some key issues of aircraft structure life control under the influence of corrosive environment [D]. Xi'an: Air Force Engineering University, 2015). We have added the following paragraph in section 4.1.

“The type of load spectrum used in this paper is the program block spectrum, the specific values are shown in Table 2, and the specific loading history is shown in Figure 4, with 1 spectrum block for every 1239 cycles. The fatigue test loading equipment is MTS-810-500kN fatigue testing machine, the loading frequency is 15Hz, the test loading dynamic load error are less than 1% of the maximum load.”

Table 4.  Program Block Spectrum

Number of program blocks

1

2

3

4

5

6

7

8

9

10

11

Fmax/kN

13.88

18.50

22.63

22.63

24.75

22.63

22.63

20.63

26.75

16.50

15.25

Fmin/kN

2.06

4.13

18.50

2.06

0.00

4.13

10.31

4.13

6.19

0.00

4.13

Cycles

305

28

12

31

3

10

202

95

3

123

427

Please see the attachment for the figures.

Figure 2.  Fatigue test loading history

  1. Response to comment:

The discussion is not extended! The obtained results must be, must be compared to the results of other articles.

Response: Thanks to the reviewer for this suggestion. Since this paper presents a new principle and method, there are no results from other literature to compare. The study of this paper is about reliability, which can be found in the literature (Shreepathi S, Guin A K, Naik S M, et al. Service life prediction of organic coatings: electrochemical impedance spectroscopy vs actual service life[J]. Journal of Coatings Technology & Research, 2011, 8(2): 191-200.) and (Li J, Wu Y Z, Li B S, et al. Estimate Method for Calendar Life of Helicopter′s Coating Based on Electrochemical Impedance [J].Equipment Environmental Engineering, 2017, (7).). The methods used are to take the average value, which means that the reliability is chosen to be 50%. The method in this paper finally determines the reliability to be around 70%, which can achieve both the safety requirements and more economical. We added the following paragraph at the end of section 4.3:

  “Since this paper presents a new principle and method, there are no results from other literature to compare. The study of this paper is about reliability, which can be found in the literature [33] and [34]. The methods used are to take the average value, which means that the reliability is chosen to be 50%. The method in this paper finally determines the reliability to be around 70%, which can achieve both the safety requirements and more economical.”

  1. Response to comment:

The SEM images must be provided for the sample after the failure of fatigue. What is fracture behavior? Ductile or brittle? What were the features? Dimple? CLeavage? What was the failure mechanism? The EDX analysis must be provided.

Response: Thanks to the reviewer for this suggestion. Based on the reviewer's suggestion, we provide the sample after the failure of fatigue as shown in Figure 1.

Please see the attachment for the figures.

Figure 1: EDS data of corrosion products and microscopic morphology of specimen fracture: (a) microscopic morphology of specimen fracture; (b) local enlarged morphology of slip band and the EDS data of area â‘ ; (c) local enlarged morphology of corrosion region at crack tip and fatigue crack growth region; (d) local enlarged morphology of fatigue crack growth region; (e) local enlarged morphology of (c); (f) local enlarged morphology of (e) and the EDS data of area â‘¡.

And the corrosion and fracture morphology characteristics were examined by scanning electron microscopy (SEM) and the composition of corrosion products was examined by energy dispersion spectrum (EDS) as shown in Figure 1. After alternating corrosion fatigue, the crack growth region of the specimen fracture was divided into three parts as shown in Figure 1(a): the first part is the corrosion area of the crack tip formed by the exposed part of the crack tip after corrosion test; the second part is the corrosion area of slip band, which is generated by the corrosion of the slip band at the crack tip along the crack growth direction during the corrosion process; The first part and the second part can be collectively referred to as the crack tip corrosion area. The third part is the fatigue growth region of the crack.

The corrosion region at the crack tip shows different surface characteristics from the fatigue growth region as shown in Figure 1(c). Because of corrosion, the crack section is dark and there are a lot of corrosion products on the surface in the corrosion area at the crack tip. From Figure 1(e), the pitting is very dense at the crack tip. And the corrosion products on the surface cracked, forming lumps with filamentous corrosion products being covered. Figure 1(f) shows that the filamentous corrosion products were composed of amorphous particles connected to each other and that the structure was loose, allowing easier absorption of water from the atmosphere. The mass percentage ratio of Al to O in area â‘¡ of Figure 1(f), resulting from the EDS analysis, was used to determine that the filamentous corrosion products and the amorphous particles were also Al(OH)3. And it also can be seen that, with the accumulation of corrosion products, the crack tip also produces secondary cracks.

The corrosion region of the slip band of crack growth fracture was observed after the alternating action of corrosion and fatigue, as shown in Figure 1(b). .e presence of Si and Ca in the EDS analysis of area â‘  indicated that corrosion mediums contained those two elements. And the obvious characteristics of the slip band can be seen as shown in Figure 1(b).

The growth region of crack would produce dimples after the alternating test of corrosion and fatigue as shown in Figure 1(d). Due to the existence of two-phase particles and high potential at the dimple, corrosion occurs first and many corrosion products are produced. As for the fatigue crack growth region, the crack growth morphology is the same as that under the pure fatigue action because it is only affected by the fatigue load.

The strength and fatigue resistance of materials are reduced after corrosion as proved by relevant investigations. Since the initial cracks of the specimens used in the test were obtained through fatigue loading, plastic region and many slip bands will be formed at the tip of the crack after the initial crack preparation is completed. When there were cracks under the action of the corrosion environment, the slip bands and plastic regions at the crack tip become anode under the corrosion environment and corroded first due to their high potential. It is different from the action of pure fatigue. Under the action of corrosion, the plastic region and slip bands of the material dissolve continuously, and many corrosion products are formed at the crack tip. With the accumulation of corrosion products, a local block cell will be formed at the crack tip, and the corrosion region of the slip band of the crack tip will be formed as shown in Figure 3(b). Compared with the crack growth region of pure fatigue, the number of slip bands generated after the alternation of corrosion fatigue will gradually disappear and decrease sharply, and many corrosion products will be formed in the original slip region.

It should be especially noted that this paper studies the reliability issue and is concerned with the fatigue life of the structure rather than fatigue fracture, crack extension, etc., so this part above is not mentioned in the article.

Special thanks to you for your good comments.

We tried our best to improve the manuscript and made some changes in the manuscript. These changes will not influence the content and framework of the paper. And we marked it in red in the revised paper.

We appreciate for your warm work earnestly, and hope that the correction will meet with approval.

Once again, thank you very much for your comments and suggestions

Looking forward to hearing from you soon.

With kindest regards,

Yours Sincerely,

Zhang Tianyu

Reviewer 4 Report

The authors should discuss what type of loading is involved in the parts. It is a known fact that fatigue loading limits the life of aluminium based parts since they do not have an endurance limit. The authors should discuss why aluminium based materials should be used and not alternate material that has an endurance limit like titanium based alloys or super alloys.

Minor changes may be required

Author Response

Dear Reviewer,

Thank you for your letter and for your comments concerning our manuscript entitled “Principle and method of determining the calendar safety life of aircraft structural protection systems” (Manuscript Number: coatings-2353657). Many thanks to you for your repeated comments and suggestions, which are greatly appreciated. Those comments are all valuable and very helpful for revising and improving our paper, as well as the important guiding significance to our research. We have studied the comments carefully and have made corrections which we hope meet with approval. Revised portions are marked in red on the paper. The main corrections in the paper and the response to your comments are as flowing:

Comment: The authors should discuss what type of loading is involved in the parts. It is a known fact that fatigue loading limits the life of aluminum based parts since they do not have an endurance limit. The authors should discuss why aluminum based materials should be used and not alternate material that has an endurance limit like titanium based alloys or super alloys.

Response: Thanks to the reviewer for this suggestion, Because the main object of this paper is the aircraft structure, most of the fuselage surfaces, landing gears and various connectors on the aircraft are aluminum alloy materials, and the aluminum alloy materials used in the current civil aircraft structure in China account for about four-fifths of all materials, so aluminum-based materials are used in this paper. We have added the following paragraph to the introduction to make it easier for the reader to understand:

“Because the main object of this paper is the aircraft structure, most of the fuselage surfaces, landing gears and various connectors on the aircraft are aluminum alloy materials, and the aluminum alloy materials used in the current civil aircraft structure in China account for about four-fifths of all materials, so aluminum-based materials are used in this paper.”

The type of load spectrum used in this paper is the program block spectrum, the specific values are shown in Table 2, and the specific loading history is shown in Figure 4, with 1 spectrum block for every 1239 cycles. The fatigue test loading equipment is MTS-810-500kN fatigue testing machine, the loading frequency is 15Hz, the test loading dynamic load error are less than 1% of the maximum load. Specific references can be found in the literature (Zhang Teng. Research on the principle and some key issues of aircraft structure life control under the influence of corrosive environment [D]. Xi'an: Air Force Engineering University, 2015). We have added the following paragraph in section 4.1.

“The type of load spectrum used in this paper is the program block spectrum, the specific values are shown in Table 2, and the specific loading history is shown in Figure 4, with 1 spectrum block for every 1239 cycles. The fatigue test loading equipment is MTS-810-500kN fatigue testing machine, the loading frequency is 15Hz, the test loading dynamic load error are less than 1% of the maximum load.”

Table 4.  Program Block Spectrum

Number of program blocks

1

2

3

4

5

6

7

8

9

10

11

Fmax/kN

13.88

18.50

22.63

22.63

24.75

22.63

22.63

20.63

26.75

16.50

15.25

Fmin/kN

2.06

4.13

18.50

2.06

0.00

4.13

10.31

4.13

6.19

0.00

4.13

Cycles

305

28

12

31

3

10

202

95

3

123

427

Please see the attachment for the figures.

Figure 2.  Fatigue test loading history

Special thanks to you for your good comments.

We tried our best to improve the manuscript and made some changes in the manuscript. These changes will not influence the content and framework of the paper. And we marked it in red in the revised paper.

We appreciate your warm work earnestly and hope that the correction will meet with approval.

Once again, thank you very much for your comments and suggestions

Looking forward to hearing from you soon.

With kindest regards,

Yours Sincerely,

Zhang Tianyu.
